# A Review about the Mycoremediation of Soil Impacted by War-like Activities: Challenges and Gaps

**DOI:** 10.3390/jof10020094

**Published:** 2024-01-24

**Authors:** Regina Geris, Marcos Malta, Luar Aguiar Soares, Lourdes Cardoso de Souza Neta, Natan Silva Pereira, Miguel Soares, Vanessa da Silva Reis, Madson de Godoi Pereira

**Affiliations:** 1Institute of Chemistry, Federal University of Bahia, Barão de Jeremoabo Street, s/n, Campus Ondina, 40170-115 Salvador, BA, Brazil; rmgeris@ufba.br (R.G.); marcosmalta@ufba.br (M.M.); miguelss@ufba.br (M.S.); 2Department of Exact and Earth Sciences, Bahia State University, Silveira Martins Street, N. 2555, Cabula, 41150-000 Salvador, BA, Brazil; luaraguiar797@gmail.com (L.A.S.); lcneta@uneb.br (L.C.d.S.N.); nspereira@uneb.br (N.S.P.); van.nessa2204@gmail.com (V.d.S.R.)

**Keywords:** fungi, bioremediation, explosives, radionuclides, toxic elements, Agent Orange

## Abstract

(1) Background: The frequency and intensity of war-like activities (war, military training, and shooting ranges) worldwide cause soil pollution by metals, metalloids, explosives, radionuclides, and herbicides. Despite this environmentally worrying scenario, soil decontamination in former war zones almost always involves incineration. Nevertheless, this practice is expensive, and its efficiency is suitable only for organic pollutants. Therefore, treating soils polluted by wars requires efficient and economically viable alternatives. In this sense, this manuscript reviews the status and knowledge gaps of mycoremediation. (2) Methods: The literature review consisted of searches on ScienceDirect and Web of Science for articles (1980 to 2023) on the mycoremediation of soils containing pollutants derived from war-like activities. (3) Results: This review highlighted that mycoremediation has many successful applications for removing all pollutants of war-like activities. However, the mycoremediation of soils in former war zones and those impacted by military training and shooting ranges is still very incipient, with most applications emphasizing explosives. (4) Conclusion: The mycoremediation of soils from conflict zones is an entirely open field of research, and the main challenge is to optimize experimental conditions on a field scale.

## 1. Introduction

Soil is necessary for life since this outermost layer of the terrestrial crust contains suitable biogeochemical characteristics to allow for the growing and nutrition of plants, thus sustaining all terrestrial food chains. Geologically, soils contain sediments derived from the intemperism (physical, chemical, and biological) of rocks and organic matter [1,2].

Naturally, soils have very high structural and compositional variability since these ecosystems have characteristics and properties that depend strongly on climatic, topographical, and chemical (related to the parent rock) aspects [3]. Figure 1 shows the global distribution of the main types of soils. Even if the classification of soils changes over time—the map above is from 2005—Figure 1 still shows the significant variability of soils according to geography. However, since humanity began systematically cultivating approximately 10,000 years ago, soils have suffered cumulative degradation [4], and the Industrial Revolution greatly intensified this worrying scenario. Consequently, it is challenging to establish direct relationships between the chemical composition of soil and its parental rock.

In terrestrial ecosystems, soils are the primary receptors for a myriad of pollutants, including polymers [5,6,7], petrochemicals [8,9,10], pharmaceuticals [11], toxic metals and metalloids [12,13,14], and pesticides [15,16]. All these pollutants come mainly from industrial, agricultural, residential, and commercial activities. However, shooting clubs, military exercises, weapon tests, and real wars, which are called war-like activities in this manuscript, comprise another type of human action responsible for the devasting environmental quality of soils [17,18,19,20]. The use of conventional, chemical, and nuclear weapons throughout the 20th and 21st centuries has released and still releases immense amounts of toxic metalloids and metals [21,22], pesticides (mainly herbicides [23,24,25,26,27]), explosives [28,29,30], and radioactive residues [31].

As military technology increased and war-like activities became global throughout the 20th century, significant land extensions received inputs of different pollutants, including toxic metals and metalloids, herbicides, and radionuclides, compromising the most noble function of soils, i.e., food production. In this sense, strategies for decontaminating soils environmentally impacted by war-like activities are essential from an environmental point of view.

Toxic metals and metalloids are commonly removed from soils by diverse chemical and physical treatments. Nevertheless, these treatments present considerable disadvantages, including high costs, inefficiency in removing accentuated concentrations of these pollutants, as well as long execution times. In turn, microbial remediation offers attractive and eco-friendly features [32,33] to treat these pollutants since suitable microorganisms in their natural or genetically modified form are present in contaminated soils [34]. For the reasons already described for the decontamination of soils containing metals and metalloids, microbial remediation also has successful applications for pesticides [35], explosives [36], and radioactive residues [37].

Soils are environments with numerous microorganisms; each gram of soil can contain more than 10 billion microbial cells. In this vast community, fungi are very representative, notably facilitating energy exchanges between the aboveground and belowground systems and increasing soil permeability. Fungi are also essential for decomposing organic residues, including cellulose, hemicellulose, and lignin biopolymers, and recycling plant nutrients [38,39]. Moreover, fungi remediate toxic substances such as metals, phenols, chlorophenol mixtures, phenanthrene, petroleum hydrocarbons, polycyclic aromatic hydrocarbons, and pesticides, among others [40,41]. In this sense, soils polluted by war-like activities comprise potentially recoverable environments via mycoremediation.

Faced with the efficiency already proven of microbial soil remediation and the worrying rate of degradation of soil ecosystems by wars (World Wars I and II, the Cold War, the Vietnam War, and the Yugoslav Civil War) and other war-like activities, this manuscript aimed to carry out a critical review (1980–2023) of the decontamination of soils impacted by these activities using mycoremediation, which is a particular type of microbial remediation.

To achieve the aim of this manuscript, the databases ScienceDirect and Web of Science were searched as the primary sources of articles on the topic of war-like soil mycoremediation, using key terms such as “fungal remediation”, “fungal bioremediation”, “mycoremediation”, “fungal degradation”, and “fungal transformation”, allied with the following search terms: “soils”, “contamination”, “pollution”, “fungi”, “fungi resistance”, “mycoremediation mechanisms”, “World War I”, “World War II”, “Vietnam War”, “Cold War”, “Yugoslav Civil War”, “soil war pollutants”, “soil weapon pollutants”, “soil warfare pollutants”, “soil bomb pollutants”, and “soil bullet pollutants.” Additional keywords, including “soil toxic metals and metalloids” (arsenic, cadmium, copper, iron, lead, nickel, and zinc), “rainbow herbicides”, “orange agent”, “2,3,7,8-TCDD”, “2,4-D (DCPA)”, “2,4,5-T (TCPA)”, “explosive”, “soil explosives”(nitroaromatics, TNT, RDX, and HMX), “soil radionuclides” (polonium, plutonium, uranium, and depleted uranium), “isotopes”, “radioactivity”, “nuclear explosion”, and “nuclear test”, were also examined in this search. To eliminate duplicate or unrelated papers, the reviewers initially checked the search results based on the title, abstract, and, subsequently, the full text of selected articles.

## 2. Theoretical Foundation

### 2.1. How Have World War I, World War II, the Cold War, the Vietnam War, and the Yugoslav Civil War Environmentally Impacted Soils?

When World War I ended on 11 November 1918, the belligerents fired approximately 1.45 billion artillery shells of different types. This immense quantity of artillery shots transferred 15 million tons of copper, iron, lead, nickel, and zinc to the soil [42]. Even 91 years after the end of World War I, Bausinger et al. [22] and Thouin et al. [43] detected very high concentrations of copper, iron, lead, and zinc in European soils.

In addition to metallic species, World War I was responsible for spreading arsenic, which is a metalloid toxic to humans. In this sense, of the approximately 1.45 billion artillery shells fired during this war, around 5% contained chemical ammunition, including arsenic compounds [42]. Bausinger et al. [22], Thouin et al. [42], and Tarvainen et al. [44] found that different European soils that were scenes of fighting in World War I still contain arsenic levels that exceed at least 1400 times the environmental limits of countries like Germany. Much of the arsenic residue in battlefield soils refers to inorganic forms of this element [14].

Once in the soils, artillery and infantry projectile shells corrode, and the metallic elements initially with zero oxidation oxidize to cations. In turn, the mobilization of these cations to plants will strongly depend on the levels of clays and humified organic matter. Regarding the arsenic deposited in the soils predominantly as oxides, the bioavailability of this element will depend on how these oxides dissolve. Because arsenic oxides are amphoteric, their dissolutions are possible over relatively wide pH ranges. The literature [45,46,47,48,49,50,51,52] discusses the toxicological effects of arsenic, lead, copper, iron, and zinc on humans.

The violent fighting during World War I also released very significant quantities of explosives, notably nitroaromatic compounds with an emphasis on 2,4,6-trinitrotoluene (TNT), whose chemical formula is C_7_H_5_N_3_O_6_ (see Figure 2). TNT has low solubility in water and does not interact appreciably with soil particles. Thus, many organisms assimilate the molecules of this compound, which can reach humans. Moreover, the chemical and microbiological decomposition of TNT in soils can occur and give rise to derivatives with very different chemical properties [53].

In a war scenario, pollution from explosives, including TNT and two other explosives (RDX and HMX) discussed below, occurs when armies abandon unexploded projectiles on the ground. In this case, the metallic capsules of these projectiles corrode over time, with the consequent release of explosives gradually into the soil. As TNT has remarkable chemical stability, this explosive stays available in the soil as the ammunition casings degrade. Approximately 90 years after the end of the Battle of Verdun, soil concentrations of TNT and several of its derivatives were found to be close to 11 mg kg^−1^ [22]. Pichtel [53] and Gao et al. [54] discuss the toxicological effects of TNT on humans in depth.

Like the soils impacted by World War I, the soils bombed during World War II also received pollution by copper, lead, zinc, and TNT. These soils continued to be widely used. However, new and more powerful explosives entered the scene during World War II, including the royal detonation explosive (RDX, hexahydro-1,3,5-trinitro-1,3,5-triazine, Figure 3A), whose chemical formula is C_3_H_6_N_6_O_6_ [55]. By 1945, the United States of America had produced 434,000 tons of RDX [56].

Like TNT, RDX is a chemically and thermally stable compound. The unexploded ordnance abandoned on the battlefield soils have acted as sources of this explosive as the capsules have degraded over time. Another explosive that emerged during World War II was the high melting explosive (HMX, octahydro-1,3,5,7-tetranitro-1,3,5,7-tetrazocine–C_4_H_8_N_8_O_8_, Figure 3B). In addition to the release of unexploded and abandoned projectiles, soil contamination by HMX occurs due to the solid and liquid residues produced during the manufacturing, transporting, and destroying of military artifacts [57]. This argument is valid for any explosive [58]. The damage that RDX and its derivatives [59,60,61] and HMX [62,63] cause to human health is vast and ranges from seizures and epilepsy to the emergence of tumors.

Similarly to metallic cations and TNT, the soil type also influences the retentions of RDX and HMX and their derivatives. Studies demonstrate that half-lives of RDX range from 60 days (silty soil) to 188 days (sandy soil). In turn, HMX presents half-lives from 40 days (sandy soils) to 2310 days (silty clay soil) [64].

A few weeks before its end, World War II inaugurated the use of nuclear weapons, and soils were one of the environmental compartments that received the most radioactive debris, containing isotopes of plutonium, cesium, and uranium [31,65].

Still in the context of the Cold War, the world also witnessed the Vietnam War, which, like other wars after World War I, made massive use of artillery fire, land mines, and highly destructive aerial bombings. Therefore, considerable portions of Vietnamese territories contain long-lasting sources of soil pollution from explosives and metals. The government of Vietnam estimates that between 6.1 and 6.6 million hectares, or 19 to 21% of Vietnam, are home to unexploded and abandoned ammunition [66]. Moreover, the Vietnam War was very peculiar in using herbicides as a military strategy. In this case, the United States of America dumped the defoliant Agent Orange over one-quarter of southern Vietnam to reveal enemy troops hiding under the tropical forests and destroy the enemy’s food crop production [23]. Over nine years, the U.S. Air Force sprayed 19 million gallons (approximately 72 million liters) of defoliants (42 million liters of Agent Orange) across the Republic of Vietnam [24,67]. The Agent Orange composition has equal parts of 2,4-dichlorophenoxyacetic acid (2,4-D) and 2,4,5-trichlorophenoxyacetic acid (2,4,5-T) and an extremely toxic by-product known as 2,3,7,8-tetrachlorodibenzeno-p-dioxin (2,3,7,8-TCDD) [24]. Like other compounds derived from war-like activities, Agent Orange causes severe damage to the human body, as described in the literature [23,68,69,70,71,72,73,74,75].

Following the ecocide logic of wars, the Yugoslav Civil War strongly impacted soils from areas subjected to intense fighting. As in any war of the 20th century [76], this war polluted soils with significant amounts of toxic and potentially toxic metals and explosives (such as TNT and RDX) from unexploded ordnance. Especially in the Yugoslav Civil War, radionuclides of uranium were available in a particular type of ammunition known as depleted uranium ammunition. For more information on how depleted uranium is obtained and why it is used as ammunition, specialized readings are recommended [77,78]. During this war, more than 30,000 projectiles containing DU (approximately 10 tons of DU) were fired on 112 sites [79,80,81], thus causing some regions to become chemically contaminated [82,83]. DU radioactivity is about 40% lower than that of natural uranium [77]. In this sense, DU toxicity is not mainly due to radioactivity but because uranium is a metal that causes damage to the kidneys, liver, bones, and brain [78].

Borgna et al. [84] concluded that attacks with DU ammunition had only punctual damage on Yugoslav soil. However, punctual concentrations of DU can spread if the filtration of uranium to groundwater occurs [85], depending on the chemical conditions of the soil [86].

### 2.2. How Do Military Training and Shooting Ranges Environmentally Impact Soils?

In 2001, North American territory contained 950 deactivated military training centers with unexploded ammunition and other types of waste from military activities [87]. By 2004, the U.S. Army estimated that 1.2 million tons of soil contained explosives and, by 2006, decommissioned military areas totaling 40,000 km^2^ in the United States of America that had unexploded ordnance [88,89].

On a smaller scale, the daily activities of civilian shooting ranges also bring environmental impacts like those of military training. Shooting ranges [90,91,92,93,94], notably, those that offer shotgun sports, are known to pollute soils with lead because each shotgun cartridge discharges up to 36 g of lead pellets over diffuse areas [84,88]. Around the world, several studies have reported impressive lead loadings (several tons) from shooting ranges deposited in soils [95,96,97,98].

The main composition of these pallets is lead in percentages that vary from 90 to 97% (m/m) [99]. J∅rgensen and Williems [95] estimated that all lead-containing pellets would undergo a chemical transformation in Danish soils over 100 to 300 years. Lin et al. [100] concluded that approximately 5% of all metallic lead contained in pellets deposited in the soils of a central shooting range in Sweden converted to lead carbonate and sulfate within 20 to 25 years. A typical round of sporting clay shooting can add 3.2 kg of lead per shooter to the soil. Therefore, many shooting ranges rival and surpass lead smelters in polluting neighboring soils [94]. Reviews [17,21] and research manuscripts [94,97] offer detailed information concerning the pollution of soils by shooting ranges. Now that we have discussed how different war-like activities pollute soil, the following section discusses the characteristics that allow fungi to clean up this ecosystem.

## 3. Fungi: Up-and-Coming Candidates to Remediate Soils Contaminated by War-like Activities

Fungi comprise the largest kingdom of higher organisms on the planet: eukaryotes with complex cell structures and abilities to make tissues and organs [101]. The estimate is that there are 2.2 to 3.8 million species of fungi on Earth, of which only 120,000 species have been isolated or described [102]. Most of these fungi are associated with soil, either within or on the soil itself or within or on various living or deceased plants and animals situated within or on the soil [101]. There can be around 20,000 km of hyphae per square meter of agricultural soil [101].

The fungal kingdom (also named Eumycota) is comprised of ten phyla, following the revision of its phylogenomic analyses by McCarthy and Fitzpatrick [103]: Cryptomycota, Microsporidia, Chytridiomycota, Monoblepharidomycota, Neocallimastigomycota, Blastocladiomycota, Zoopagomycota, Mucoromycota, Ascomycota, and Basidiomycota. The last two phyla are combined in the subkingdom Dikarya, each of which houses about 64,000 and 32,000 known species, respectively [101,103].

Fungi have a heterotrophic mode of nutrition, obtaining their nutrients by enzymatic extracellular digestion from a range of complex materials followed by absorption of the solubilized breakdown products. In their life cycle, spores of both sexual and asexual reproduction act as the prime units of dispersal in filamentous fungi, from which one or more germ tubes (young hyphal tips) emerge, forming long cylindrical cells known as hyphae and extending away from the spore in a typical apically polarized manner to develop the mycelium network (Figure 4); this requires a progressive supply of proteins, lipids, and cell wall precursors to the hyphal tip [101,104]. Moreover, the capture of these organic matter present in the substrate follows the exudation of enzymes capable of catalyzing the breakdown of carbohydrates and proteins into peptides, amino acids, and monosaccharides–molecules small enough to be absorbed through diffusion into the cell wall. As the mycelium grows, it can develop fruiting bodies that are involved in sexual reproduction for the production and dispersal of new spores [105,106].

Hyphae filaments have a rigid, complex cell wall and moving protoplasm (cytosol) divided into compartments by cross walls termed septa, allowing cellular components to move through these [107]. The plasma membrane comprises a phospholipid bilayer associated with transmembrane proteins and ergosterol and some enzymes such as integral membrane proteins chitin synthase and glucan synthase [104,107]. Many excellent review articles in the literature describe and discuss cell wall structures and functions in detail, such as protective barriers against other microorganisms and osmotic lysis, the binding site for enzymes to degrade macromolecules into their monomers, and signaling [107,108,109]. However, it is essential to know the composition of the fungal cell wall and how the hyphae are elongated to understand the process of mycoremediation. Fungal cell walls are typically composed of glucans (50–60%, including *β*-1,3-, mixed *β*-1,3-/*β*-1,4-, *β*-1,6-, and *α*-1,3-glucans), chitin (10–20%), mannans and/or galactomannans, and glycoproteins (20–30%) [108,110]. Proteins, including “integral cell wall proteins”, e.g., glycoproteins (mannoproteins, galactomannan proteins, and glycosylphosphatidylinositol–GPI anchor), and “nonintegral cell wall proteins” (heat-shock, glycolytic enzymes, and hydrophobins) have been estimated to account for approximately 30–50% of yeast and 20–30% of cell walls of filamentous fungi [108,109,110].

Three regions constitute the fungal wall: (1) the inner layer placed over the cell membrane is constituted by chitin and *α*-1,3-glucan, forming a tightly packed, rigid, and hydrophobic scaffold; (2) the outermost layer consists of a highly mobile shell of mannoproteins and *α*-1,3-glucan; and (3) a well-hydrated intermediate region comprises *β*-1,3, *β*-1,4, and *β*-1,6-glucans embedding the hydrophobic core (Figure 4) [111].

### 3.1. Mycoremediation and Its Techniques

“*When looking for nature-based solutions to some of our most critical global challenges, fungi could provide many of the answers*.” (State of the World’s Fungi 2018 by Katherine Willis, Director of Science, Royal Botanic Gardens, Kew)

Bioremediation, by definition, refers to the cost-effective and environmentally friendly method for the efficient conversion of xenobiotics (toxic and recalcitrant pollutants) into environmentally benign products through the action of natural biological treatments of polluted systems such as land and water [112]. Biological agents such as animals, plants, fungi, bacteria, and other organisms, whether naturally occurring, adapted, or modified, have their biochemical capabilities directed toward the removal or transformation/attenuation of environmental pollutants [113,114].

Fungi and their morphologic, physiological, and metabolic characteristics are involved in the conversions of organic and inorganic compounds. Fungal degradative activities (also named mycotransformation or mycodegradation) have been recognized in many circumstances when these microorganisms break down different types of wood, stored paper, textiles, plastics, leather, electro-insulating materials, and various wrapping materials [115]. Therefore, mycoremediation is the bioremediation division that employs fungi to degrade, restore, and heal contaminated ecosystems [112,115].

The substantial contribution of these living organisms to various fields of biotechnology can be tracked through records obtained via bibliographic surveys employing the term “fungal biotechnology”, with 13,187 publications distributed across categories in the Web of Science, including *Biotechnology Applied Microbiology*, *Environmental Sciences*, and *Soil Science*. When employing the terms “fungal remediation”, “fungal bioremediation”, and “mycoremediation”, directing attention to fungi in pollutant remediation, it is possible to observe the increasing trend in publications since the 1990s as an eco-friendly and practical approach (Figure 5). Thus, the use of the term “mycoremediation” in research on bioremediation is still in its emerging stages. Yet, it mirrors the upward trajectory in utilizing these microorganisms to remediate polluted environments.

Despite the magnitude of 120,000 fungi found globally, only a few species have been associated with mycoremediation [115,116]. Principal genera of fungi include *Aspergillus, Cryptococcus, Curvularia, Drechslera, Fusarium, Lasiodiplodia, Mucor, Penicillium, Rhizopus*, and *Trichoderma* [116]. White-rot fungi, including *Phanerochaete chrysosporium*, *Trametes versicolor*, *Bjerkandera adjusta,* and *Pleurotus* sp., are also main agents of the biodegradation of lignininous material and have demonstrated bioremediation potential by different ligninolytic enzyme actions [112,115]. Moreover, due to lifestyle conditions, marine, extremophilic, and symbiotic fungi (mycorrhiza and endophytes) are potential candidates for diverse bioremediation applications [116,117,118].

Fungi can be employed to promote treatment in several matrices polluted with polycyclic aromatic hydrocarbons (PAHs) [119], petroleum hydrocarbons [120,121,122,123,124], biphenyls [125], phthalates [126], polychlorinated herbicides such as polychlorinated dibenzo-p-dioxins–PCDD [127], chlorinated insecticides and pesticides [128], textiles dyes [129], pharmaceutical substances like antibiotics sulfonamides [130], and norfloxacin [131]. Toxic metals, including cadmium, copper, mercury, lead, manganese, nickel, zinc, and iron, and metalloids, with an emphasis on arsenic, are extensively used in different types of industries, being released in high amounts to their effluents, causing direct or indirect environmental contamination where fungi can act to remediate [132,133,134,135,136]. Excellent books and original and review articles describe how these pollutants have been remediated, and some of them are documented in this manuscript as recommended readings, given the specific nature of the treatment for each type of wastewater [112,115,116,117,135,137,138,139,140].

Two primary positive considerations can be observed to support the use of fungi in the remediation of environmental contaminants as opposed to traditional physicochemical processes, namely, they are low-cost and environmentally acceptable, in line with principles of green chemistry [141]. Filamentous fungi can produce a large amount of biomass during their growth and development, which is cheap and readily available as a byproduct of several essential economic fermentation processes [142]. This robust growth of fungus produces a vast hyphal network with a high surface area-to-volume ratio, extracellular ligninolytic enzyme production, resistance to toxic metals, and adaptability to pH and temperature variations, endorsing their use [117].

Several bioremediation technologies are employed to remove or stabilize pollutants from contaminated environments by either applying bioremediation at the point of contamination—named “in situ” bioremediation—or transferring contaminated material (soils, for example) to a remote treatment facility—the “ex situ” methodology [116]. In situ processes include biosparging, biostimulation, bioaugmentation, bioventing, natural attenuation, bioslurping, bioleaching, phytoremediation, and mycoremediation techniques and ex situ methods comprise biopiling, composting, land farming, the use of biofilters, and the use of and bioreactors [116,132]. However, the understanding adopted in this manuscript is that mycoremediation and phytoremediation are used in not just one of the bioremediation techniques but indeed are rather expansive as they play a fundamental role in most of the bioremediation techniques, being applied isolated or in a combination as biological agents of remediation. Thus, fungi are considered as one of the biological agents that perform remediation. Figure 6 introduces a flowchart of how fungi perform through their mechanisms in these different bioremediation techniques.

Intrinsic bioremediation and biostimulation are based on optimizing the local microbiota’s metabolism. The former is a natural attenuation process conducted in situ using the inherent propensities of the local microbial population to convert environmental pollutants into nontoxic forms without human intervention [143,144]. Biostimulation exploits the intrinsic ability of local microbiota to succeed in contaminated environments by adding nutrients (carbon, phosphorus, nitrogen) and optimizing environmental conditions (temperature, pH). These conditions improve the growth rate of degrading microorganisms, increasing biomass and expanding their metabolic activity [145,146].

Bioaugmentation is a technique that uses specific microorganisms with high resistance and a good capacity to degrade specific contaminants rapidly. They may be a microorganisms with or without genetic modification, and they are introduced into the treatment area as a pre-adapted pure culture or consortium [143,147].

Biosparging, bioventing, and bioslurping techniques aim to enhance physicochemical conditions to intensify the bioremediation process of the indigenous microbiota. In biosparging, volatile components migrate to the unsaturated zone when air is injected and undergo biodegradation. In bioventing, the stimulation of airflow is regulated. In bioslurping, bioremediation occurs through a combination of bioventing with vacuum-enhanced pumping [116].

On the other hand, ex situ remediation’s fundamental goal is to introduce suitable nutrients and humidity and oxygen conditions to native microorganisms using five techniques: composting, biopiling, land farming, using a biofilter, and using a bioreactor. During composting, contaminated soil is placed in treatment containers, mixed with biomass (compost), and aerated during composting by combining the elements for a few weeks [127]. Composting is an age-old strategy for remediating soils contaminated with organic pollutants, such as PAHs, by employing lignin-degrading fungi, including the white-rot fungus *Phanerochaete sordida* [148]. This type of bioremediation can be combined with bioaugmentation to remediate soil and sediment metals. Huang and colleagues [149] investigated the impact of inoculating the fungus *Phanerochaete chrysosporium* on Pb’s bioavailability and the bacterial community’s diversity in a compost pile of agricultural residues contaminated with this metal.

In the biopiling technique, the soil is arranged in piles to enable the control of leachate from these piles, involving the addition of nutrients or minerals and the injection of air. This method remediates soils contaminated with petroleum hydrocarbons, PAHs, polychlorinated biphenyls (PCBs), and pharmaceutical wastes [143,150]. The basis of landfarming is the controlled application of waste on a soil surface to allow the natural microbiota to biodegrade the contaminants aerobically, such as drilling wastes and refinery waste materials. It involves placing contaminated material, aeration, watering, fertilizing, and removing treated material [143].

The soil slurry reactor process involves mixing contaminated water, soil, and other additives (essential nutrients and oxygen) in a bioreactor to allow the native microorganisms to break down the pollutants [116]. However, this methodology is not famous for contaminated land bioremediation since large amounts of water are required [143].

Therefore, in all bioremediation techniques, one can observe the considerable activity of the native microbiota or the addition of microorganisms capable of rapidly degrading contaminants within their matrices. Fungi are among these microbial agents involved in all bioremediation processes. Mycoremediation mechanisms are complex and offer varied possibilities to reduce the bioavailability of pollutants or degrade them. The following section describes these mechanisms, emphasizing fungi’s relevance to decontaminating soils impacted by war-like activities.

### 3.2. Mycoremediation Mechanisms

Regardless of the bioremediation technique employed to treat pollutants, fungi can partially transform or break down the contaminants into simpler forms and utilize them as substrates for their growth by operating through multiple mechanisms distributed into two categories: biosorption and bioaccumulation. These mechanisms are not discussed in detail here as several mechanisms studies have been conducted, and the literature is already available in previous sections of this review.

Biosorption, first introduced in 1951, is defined as the removal/binding of different kinds of organic compounds (such as organic solvents, synthetic dyes, herbicides, insecticides, and pesticides) and inorganic materials (like metals, metalloids, and radionuclides) in their soluble or insoluble forms from an aqueous solution [151,152,153]. The term sorption includes absorption and adsorption: absorption is the inclusion of a substance in one form into another different form, while adsorption is the physical attachment of molecules/ions on a solid material’s surface by bonding [154].

In this context, the remediation process for pollutants by fungi can begin through physicochemical interactions between the toxic agents (biosorbate) and the fungal cell wall surface (biosorbent). These interactions depend on the biosorbate’s properties, such as solubility, molecular or ionic size, reactivity, and hydrophobicity, and the biosorbent's surface charge and chemical composition [155]. Typically, fungal cell walls are composed of glucans, chitin, mannans, galactomannans, glycoproteins (mannoproteins, galactomannan proteins, and glycosylphosphatidylinositol–GPI anchor), “nonintegral cell wall proteins” (heat-shock, glycolytic enzymes, and hydrophobins), and lipids [141]. These biocompounds comprise numerous functional groups, like the carboxyl (-COOH), carboxylate (-COO^−^), hydroxyl (-OH), amino (-NH_2_), thiol (-SH), methoxy (-OCH_3_), phosphate (-OPO_3_^−^), and sulfate (-OSO_3_^−^) groups, as well as esters (-COOR) and amides (-CONH_2_). These functional groups play a pivotal role in biosorption through adsorption by engaging in electrostatic interactions and van der Waals forces.

#### 3.2.1. Fungal Biosorption

Although living organisms invariably perform biosorption, the term “biosorption” is primarily employed to denote bioremediation using dead/inactive biomass. It is described as a physicochemical event characterized by a passive and metabolically independent process that occurs rapidly, akin to conventional adsorptive or ion exchange methods [156]. Mycosorption results from the attraction of functional groups on the fungal cell wall, which occurs through a physical or chemical process and depends on the fungal biomass [116]. *Phialomyces macrosporus*, for example, significantly reduced Cd and Pb from synthetic aqueous media by biosorption mechanisms, with a reduction of more than 80% [157].

The biosorption process by inactive biomass can occur through various mechanisms such as physical adsorption, precipitation, ion exchange, and complexation [116,158]:(a)Physical adsorption: functional groups in the cell wall interact electrostatically and through van der Waals forces with pollutants.(b)Precipitation: precipitation or solidification is the process of transforming, for example, the toxic metal compounds into their precipitate form, which is less poisonous and almost negligible [159].(c)Ion exchange: based on the ion exchange mechanism between the sorbent and the studied pollutants through the replacement (exchange) of protons from the exchangeable sites present on the biosorbent surface with contaminants (e.g., metal ions); this mechanism is facilitated by the existence of hydroxyl, carboxyl, and phenols groups [155].(d)Complexation: functional groups in the cell wall provide the ligand atoms necessary to form complexes with metal ions, which attract and retain metals in the biomass [116]. The formation of surface complexes involves the interaction of pollutants (e.g., metal ions) with oxygen donor atoms from the oxygen-containing functional groups (coordination) [116].

#### 3.2.2. Fungal Bioaccumulation

Bioaccumulation can be understood as a biotransformation process involving an active organism’s retention and concentration of a substance or metal [160]. In the case of metals, bioaccumulation occurs as the metal is transported across the membrane into the cell’s cytoplasm, where the sequestered metal becomes immobilized within the cell. It is a complex process divided into two stages. As previously mentioned, biosorption occurs in the first stage of rapid chelation of the metal to the cell wall, which is independent of metabolism. The second stage is characterized by the active transportation of metal ions into cells across the cell membrane [160,161].

The mechanism for pollutant removal using live fungal cells, which is metabolism-dependent, can occur extracellularly—on the cell wall surface—or intracellularly. Bioaccumulation mechanisms involve enzymatic and non-enzymatic processes in both extracellular and intracellular modes. The Fungal Biodegradation and Biotransformation section will briefly discuss the enzymatic processes.

The non-enzymatic processes include accumulation inside the cell via active (transport systems) and/or passive (diffusion) uptake mechanisms, exclusion by a permeability barrier, adsorption on extracellular structures (cell wall, capsule, slime), extra- and intracellular precipitation, efflux pumps, and the chelation of metal and metalloids [135]. Fungal biomineralization or bioprecipitation counteract the toxic compounds by oxalate production, a critical metabolite that plays a significant role in many metal and mineral transformations mediated by fungi [162].

In bioaccumulation, metallic species can bind to proteins, leading to precipitation and insertion into specific cellular structures or organelles [163,164,165,166]. Cells form complexes with the undesired metal and sequester it within intracellular organelles for eventual export through efflux systems. Toxic effects can occur, such as the deterioration of biomolecules, leading to changes in cellular properties.

##### Fungal Biodegradation and Biotransformation

Fungi are notably recognized for their decomposer roles with many extracellular and intracellular enzymes, which can be included in biodegradation and biotransformation strategies for remediating polluted areas. Biodegradation happens when filamentous fungi produce extracellular, non-specific, lignin-modifying enzymes, laccases, peroxidases, cellulases, xylanases, amylases, proteases, lipases, and catalases with the capacity to transform pesticides, dyes, and other organic compounds by hydrolyzing polymeric compounds [112,124,162,167]. Inside the cell, intracellular enzymes catalyze diverse reactions, resulting in the biotransformation of toxic compounds, including chemical reactions such as oxidation (cytochrome P 450–CYP 450 monooxygenases, glutathione transferases, oxidative coupling products, hydroxylation), reduction, hydrolysis, alkylation, dehalogenation, transferases (S-transferase, GST) CO_2_ emission, and others [162,168]. The microorganisms degrade these compounds with the help of endo- and exoenzymes by utilizing them as carbon sources, clearing the toxic compounds from the environment [169].

White-rot fungi are ligninolytic fungi with extensive branching, enabling them to spread in the environment and efficiently access pollutants. They secrete various extracellular ligninolytic enzymes, such as laccase, lignin peroxidase, and manganese peroxidase, that are used not only for the degradation of lignin but also for other pollutants [112,117,162]. For example, a white-rot fungus, *Trametes versicolor*, has been used to treat effluents from the industries of pulp and paper, food, textile, biofuel, cosmetics, and synthetic chemistry [169]. *Phanerochaete chrysosporium*, a white-rot fungus, is acknowledged for remediating organic pollutants [170,171], and *Phanerochaete sordida* was used to treat creosote for contaminated soil [148]. Here, fungi’s potential remediation mechanisms to address environmental pollutants are briefly described. These are well substantiated by the various scientific articles supporting this manuscript. Figure 7 is an overview of the perspective of this manuscript on the various pathways through which fungi play a role in remediation.

Table 1 and Appendix A list fungi used to remove explosives, metalloids, radionuclides, and herbicides from different media (liquid cultures and/or soil). There is considerable taxonomic diversity, although the taxa deleting specific pollutants are phylogenetically related. Microfungi involved in the bioremediation of explosives, herbicides, and radionuclides are found predominantly in Ascomycetes.

Concerning metals that have contaminated soils due to war-like activities, the primary problems are the contents of lead, nickel, copper, and zinc. Due to the many publications on the mycoremediation of toxic metals and metalloids, some excellent reviews [132,135,172,173,174] are cited.

For the pollutants metals and metalloids, radionuclides, and herbicides, this bibliographic survey (1980–2023) did not identify any publications involving soil impacted by war-like activities. Contrarily, the literature reports publications (some works listed in Table 1 and Appendix A) concerning the mycoremediation of soil containing explosives after war-like activities. A much more extensive citation about the mycoremediation of soils impacted by war-like activities and containing explosives is in Section 4.

As can be noted in Table 1 and Appendix A, living fungal cells have stood out in the mycoremediation of toxic metalloids, radionuclides, herbicides, and explosives. The preference for living fungi highlights the environmental advantages of degrading the structure of organic pollutants (explosives and herbicides) instead of transferring them to another phase, as in purely adsorptive processes. Even concerning metalloids and radionuclides, which cannot be degraded, living fungi can effectively and safely immobilize these pollutants in environments such as soil. Additionally, it is essential to note that, with or without the possibility of pollutant degradation, mycoremediation with living cells presents a removal efficiency greater than 60% for most of the applications listed in Table 1 and Appendix A. Despite these advantages, adequate nutritional conditions must be present in the media subjected to mycoremediation. Otherwise, the fungi will not efficiently decompose or immobilize pollutants.

**Table 1 jof-10-00094-t001:** Uses of fungi to remove explosives, metalloids, radionuclides, and herbicides from different media.

Pollutants/Fungal Species	Remediation Techniques and Mechanisms *	Treatment	Removal	Ref.
**RDX/White-rot fungi**				
*Phanerochaete chrysosporium*	Biodegradation Biomineralization	1.25 nmoles 1.25 nmoles	66.6 ± 3.2% 76 ± 3.9%	[175,176]
*Phanerochaete chrysosporium*	Biodegradation Biomineralization (disappearance)	100 μg mL^−1^	22%	[176,177]
*Cyatus pallidum*	Biodegradation Biomineralization (disappearance)	100 μg mL^−1^	21%	[177]
**RDX/Micromycetous fungi**				
*Cunninghamella echinulate*	Biodegradation Biomineralization (disappearance)	100 μg mL^−1^	12%	[177]
*Cladosporium resinae*	Biodegradation Biomineralization (disappearance)	100 μg mL^−1^	31%	[177]
**TNT/Wood-decaying basidiomycetes**				
*Fomes fomentarius*	Biodegradation Biomineralization	250 μΜ (56.9 ppm)	93%	[178]
*Hypholoma fasciculare*	Biodegradation Biomineralization	250 μΜ (56.9 ppm)	96%	[178]
*Kuehneromyces mutabilis*	Biodegradation Biomineralization	250 μΜ (56.9 ppm)	100%	[178]
*Laetiporus sulphureus*	Biodegradation Biomineralization	250 μΜ (56.9 ppm)	100%	[178]
*Lentinula edodes*	Biodegradation Biomineralization	250 μΜ (56.9 ppm)	90%	[178]
*Panus tigrinus*	Biodegradation Biomineralization	250 μΜ (56.9 ppm)	87%	[178]
*Phellinus robustus*	Biodegradation Biomineralization	250 μΜ (56.9 ppm)	78%	[178]
*Pleurotus abellatus*	Biodegradation Biomineralization	250 μΜ (56.9 ppm)	100%	[178]
*Pleurotus ostreatus*	Biodegradation Biomineralization	250 μΜ (56.9 ppm)	100%	[178]
*Trametes suaveolens*	Biodegradation Biomineralization	250 μΜ (56.9 ppm)	100%	[178]
*Trametes versicolor*	Biodegradation Biomineralization	250 μΜ (56.9 ppm)	100%	[178]
*Trametes versicolor* *Sclerotium rolfsii*	Biotransformation	50 mg L^−1^	66% 82%	[179]
**TNT/Litter-decaying basidiomycetes**				
*Agaricus eastivalis*	Biodegradation Biomineralization	250 μΜ (56.9 ppm)	100%	[178]
*Agaricus bisporus*	Biodegradation Biomineralization	250 μΜ (56.9 ppm)	100%	[178]
*Agrocybe aegerita*	Biodegradation Biomineralization	250 μΜ (56.9 ppm)	100%	[178]
*Agrocybe praecox*	Biodegradation Biomineralization	250 μΜ (56.9 ppm)	100%	[178]
*Clitocybe odora*	Biodegradation Biomineralization	250 μΜ (56.9 ppm)	88%	[178]
*Coprinus comatus*	Biodegradation Biomineralization	250 μΜ (56.9 ppm)	82%	[178]
*Lepista nebularis*	Biodegradation Biomineralization	250 μΜ (56.9 ppm)	94%	[178]
*Paxillus involutus*	Biodegradation Biomineralization	250 μΜ (56.9 ppm)	100%	[178]
*Stropharia rugosoannulata*	Biodegradation Biomineralization	250 μΜ (56.9 ppm)	100%	[178]
*Stropharia rugosoannulata*	Biotransformation Biomineralization	50 μΜ	UD	[180,181]
**TNT/White-rot fungi**				
*Bjerkandera adusta*	Biotransformation Biomineralization	87 μΜ	39.7%	[182]
*Cyathus stercoreus*	Biodegradation Biomineralization	90 mg L^−1^	67%	[183]
*Gymnopilus luteofolius*	Biotransformation	100 mg kg^−1^	54 ± 24%	[184]
*Irpex lacteus*	Biodegradation Biomineralization	50 mg L^−1^	100%	[185]
*Nematoloma frowardii*	Biotransformation Biomineralization	50 μΜ	UD	[180,181]
*Phanerochaete chrysosporium*	Biodegradation Biomineralization	50 mg L^−1^	91.4%	[185]
*Phanerochaete chrysosporium*	Biodegradation Biomineralization	1.25 nmoles 57.9 nmoles	50.8 ± 3.2% 6.3 ± 3.9%	[175,176]
*Phanerochaete chrysosporium*	Biodegradation Biomineralization	90 mg L^−1^	94%	[183]
*Phanerochaete sordida*	Biodegradation Biomineralization	90 mg L^−1^	90%	[183]
*Phanerochaete velutina*	Biotransformation	100 mg kg^−1^	80 ± 4%	[184]
*Phlebia brevispora*	Biodegradation Biomineralization	90 mg L^−1^	87%	[183]
*Pleurotus ostreatus*	Biodegradation Biomineralization	50 mg L^−1^	100%	[185]
*Pycnoporus coccineus*	Biodegradation Biomineralization	50 mg L^−1^	100%	[185]
*Schizophyllum commune*	Biodegradation Biomineralization	50 mg L^−1^	100%	[185]
**TNT/Micromycetous fungi**				
*Alternaria* sp.	Biodegradation Biomineralization	250 μΜ (56.9 ppm)	74%	[178]
*Aspergillus niger*	Biodegradation Biomineralization	250 μΜ (56.9 ppm)	90%	[178]
*Aspergillus niger*	Biodegradation Bioaugmentation	200 mg kg^−1^	15% 80%	[186]
*Aspergillus terreus*	Biodegradation Biomineralization	250 μΜ (56.9 ppm)	100%	[178]
*Aspergillus* sp.	Biodegradation	68 mg L^−1^	44%	[187]
*Cunninghamella elegans*	Biodegradation Biomineralization	250 μΜ (56.9 ppm)	100%	[178]
*Fusarium oxysporum*	Biodegradation Biomineralization	250 μΜ (56.9 ppm)	100%	[178]
*Fusarium solani*	Biodegradation Biomineralization	250 μΜ (56.9 ppm)	96%	[178]
*Fusarium* sp.	Biodegradation Biomineralization	250 μΜ (56.9 ppm)	100%	[178]
*Mucor mucedo*	Biodegradation Biomineralization	250 μΜ (56.9 ppm)	95%	[178]
*Mucor* sp.	Biodegradation Biotransformation	200 mg L^−1^	39%	[186]
*Mucor* sp.	Biodegradation Bioaugmentation	200 mg kg^−1^	21% 80%	[186]
*Neurospora crassa*	Biodegradation Biomineralization	250 μΜ (56.9 ppm)	100%	[178]
*Penicillium frequentans*	Biodegradation Biomineralization	250 μΜ (56.9 ppm)	100%	[178]
*Penicillium* sp.	Biodegradation Biomineralization	250 μΜ (56.9 ppm)	100%	[178]
*Rhizoctonia solani*	Biodegradation Biomineralization	250 μΜ (56.9 ppm)	90%	[178]
*Rhizopus nigricans*	Biomineralization	100 mg L^−1^	Almost 100%	[176]
*Thermomyces lanuginose*	Biotransformation Bioreduction	1.5% (w w^−1^)	7.8%	[176,188]
*Trichoderma* viride	Biodegradation Biotransformation	200 mg L^−1^	42%	[186]
*Trichoderma viride*	Biotransformation	50 and 100 ppm	UD	[189]
*Trichotecium* sp.	Biodegradation Biotransformation	200 mg L^−1^	40%	[186]
**Plutonium/White-rot fungi**				
*Pleurotus ostreatus*	Bioaccumulation Uptake	UD	UD	[190]
**Uranium/Micromycetous fungi**				
*Aphanocladium spectabilis*	Biosorption (using dead biomass)	300 mg L^−1^	54.03%	[191]
*Acremonium minutisporum*	Biosorption (using dead biomass)	300 mg L^−1^	53.83%	[191]
*Aspergillus niger*	Bioaccumulaton Bioprecipitation	UD	UD	[192]
*Gongronella butleri*	Biosorption (using live biomass)	100 mg L^−1^	UD	[193]
*Paecilomyces javanicus*	Bioaccumulaton Bioprecipitation	UD	UD	[192]
*Penicillium piscarium, Penicillium citrinum, Penicillium ludwigii*	Biosorption (using live biomass)	100 mg L^−1^	UD	[193]
*Penicillium piscarium*	Biosorption (using dead biomass)	100 mg L^−1^	97.1% 92.2%	[194]
*Talaromyces amestolkiae*	Biosorption (using live biomass)	100 mg L^−1^	UD	[193]
**2,4-D/White-rot fungi**				
*Pleurotus ostreatus*	Bioaccumulation Biotransformation	53 g L^−1^	99.3%	[195]
**2,4-D/Micromycetous fungi**				
*Aspergillus penicilloides*	Bioaccumulation Biodegradation	100 mg L^−1^	52%	[196]
*Emericella nidulans*	Biosorption (dead biomass) Biosorption (live biomass) Adsorption and uptake	0.12 mM 0.1 mM	70% 75%	[197]
*Eupenicillium* spp.	Biodegradation	100 mg L^−1^	26%	[198]
*Fusarium* sp.	Bioaccumulation Biodegradation	200 mg L^−1^	50%	[199]
*Mortierella isabellina*	Bioaccumulation Biodegradation	100 mg L^−1^	46%	[196]
*Penicillium miczynskii*	Biosorption (using dead biomass) Biosorption (using live biomass) Adsorption and uptake	0.12 mM 0.1 mM	40% 75%	[197]
*Penicillium chrysogenum*	Bioaccumulation Biodegradation	600 mg L^−1^	71.34%	[200]
*Penicillium chrysogenum*	Bioaccumulation Biodegradation	100 mg L^−1^	25%	[201]
*Rhizopus stolonifer*	Bioaccumulation Biodegradation	600 mg L^−1^	47.87%	[200]
*Rigidoporus* sp.	Bioaccumulation Biodegradation	200 mg L^−1^	100%	[199]
*Talaromyces* spp.	Biodegradation	100 mg L^−1^	3%	[198]
*Trichoderma koningii*	Bioaccumulation Biodegradation	600 mg L^−1^	52.82%	[200]
*Trichoderma viride*	Bioaccumulation Biodegradation	600 mg L^−1^	59.47%	[200]
*Umbelopsis isabellina*	Bioaccumulation Biodegradation	0.11 mM	98%	[202]
*Verticillium* sp.	Bioaccumulation Biodegradation	200 mg L^−1^	80%	[199]
**2,4,5-T/Micromycetous fungi**				
*Eupenicillium* sp. VN 5-2-2-	Biodegradation	100 mg L^−1^	8%	[198]
*Eupenicillium* sp. VN 10-2-2-	Biodegradation	100 mg L^−1^	13%	[198]
*Fusarium* sp.	Bioaccumulation Biodegradation	200 mg L^−1^	50%	[199]
*Rigidoporus* sp.	Bioaccumulation Biodegradation	200 mg L^−1^	100%	[199]
*Verticillium* sp.	Bioaccumulation Biodegradation	200 mg L^−1^	70%	[199]
**TCDD/White-rot fungi**				
*Rigidoporus* sp.	Bioaccumulation Biotransformation	0.5	73%	[203]
**As/Micromycetous fungi**				
*Absidia spinosa*	Bioaccumulation	50 mg L^−1^	115 μg g^−1^	[204]
*Acidomyces acidophilus*	Biosorption uptake	100 mg L^−1^	70.3%	[205]
*Arthroderma benhsmiae*	Bioaccumulation Biosorption (B) Biovolatilization (V)	10 mg L^−1^	B: 0.218 g kg^−1^ V: 5.21 mg kg^−1^	[206]
*Aspergillus clavatus*	Bioaccumulation Biovolatilization	5 mg L^−1^	20%	[207]
*Aspergillus niger* A	Bioaccumulation Biovolatilization	5 mg L^−1^	26.8%	[207]
*Aspergillus niger* B	Bioaccumulation Biovolatilization	5 mg L^−1^	9.2%	[207]
*Aspergillus flavus*	Biosorption Biovolatilization	0.25 mg 0.05 mg	0.015 mg (0.068 mg)	[208]
*Aspergillus nidulans*	Bioaccumulation Biosorption (B) Biovolatilization (V)	10 mg L^−1^	B: 0.190 g kg^−1^ V: 4.62 mg kg^−1^	[206]
*Aspergillus niger* *Aspergillus* spp.	Bioaccumulation Biosorption	250 mM	53.92% 52.54%	[209]
*Aspergillus oryzae*	Bioaccumulation Biosorption (B) Biovolatilization (V)	10 mg L^−1^	B: 0.250 g kg^−1^ V: 6.4 mg kg^−1^	[206]
*Aspergillus ustus* *Aspergillus* sp.	Biosorption uptake	10 ppm	80% 56%	[210]
*Cephalotrichum nanum*	Bioaccumulation	50 mg L^−1^	218 μg g^−1^	[204]
*Emericella* sp.	Bioaccumulation Biosorption (B) Biovolatilization (V)	10 mg L^−1^	B: 0.179 g kg^−1^ V: 3.62 mg kg^−1^	[206]
*Eupenicillium cinnamopurpureum*	Biosorption Biovolatilization	0.25 mg 0.05 mg	0.023 mg (0.028 mg)	[208]
*Fusarium oxysporum*	Bioaccumulation Biovolatilization	40 mg L^−1^	13.65 μg g^−1^ day^−1^ (46.35 μg g^−1^ day^−1^)	[211]
*Fusarium oxysporum*	Bioaccumulation Biotransformation	50 mg L^−1^	UD	[212]
*Fusarium* sp.	Bioaccumulation Biosorption (B) Biovolatilization (V)	10 mg L^−1^	B: 0.258 g kg^−1^ V: 6.15 mg kg^−1^	[206]
*Metarhizium marquandii*	Bioaccumulation	50 mg L^−1^	129 μg g^−1^	[204]
*Neosartorya fischeri*	Biosorption Biovolatilization	0.25 mg 0.05 mg	0.003 mg (0.180 mg)	[208]
*Neocosmospora* sp.	Bioaccumulation Biovolatilization	10 mg L^−1^	57.82%	[213]
*Penicillium janthinellum*	Bioaccumulation Biovolatilization	40 mg L^−1^	13.67 μg g^−1^ day^−1^ (54.34 μg g^−1^ day^−1^)	[211]
*Penicillium janthinellum*	Bioaccumulation Biotransformation	50 mg L^−1^	UD	[212]
*Penicillium glabrum*	Bioaccumulation Biovolatilization	5 mg L^−1^	25.2%	[207]
*Penicillium* sp.	Bioaccumulation Biovolatilization	10 mg L^−1^	58.38%	[213]
*Purpureocillium lilacinum*	Bioaccumulation	50 mg L^−1^	133 μg g^−1^	[204]
*Rhizomucor variabilis*	Bioaccumulation Biosorption (B) Biovolatilization (V)	10 mg L^−1^	B: 0.185 g kg^−1^ V: 3.63 mg kg^−1^	[206]
*Rhizopus* sp.	Bioaccumulation Biovolatilization	10 mg L^−1^	60.21%	[213]
Sterile mycelial strain FA-13	Bioaccumulation Biovolatilization	10 mg L^−1^	65.81%	[213]
*Talaromyces wortmannii*	Biosorption Biovolatilization	0.25 mg 0.05 mg	0.029 mg (0.027 mg)	[208]
*Talaromyces flavus*	Biosorption Biovolatilization	0.25 mg 0.05 mg	0.025 mg (0.025 mg)	[208]
*Talaromyces* sp.	Biosorption (using dead biomass) Biosorption (using live biomass) Adsorption and uptake	50 mg L^−1^	As(III): 5.24% As(v): 26.2%	[214]
*Trichoderma asperellum*	Bioaccumulation Biovolatilization	40 mg L^−1^	56.02 μg g^−1^ day^−1^ (51.87 μg g^−1^ day^−1^)	[211]
*Trichoderma asperellum*	Bioaccumulation Biotransformation	50 mg L^−1^	UD	[212]
*Trichoderma atroviride*	Bioaccumulation Biovolatilization	1 g L^−1^	70%	[215]
*Trichoderma* sp.	Bioaccumulation Biovolatilization	10 mg L^−1^	UD	[213]
*Trichophyton verrucosum*	Bioaccumulation Biosorption (B) Biovolatilization (V)	10 mg L^−1^	B: 0.205 g kg^−1^ V: 5.03 mg kg^−1^	[206]
*Trichoderma viride*	Bioaccumulation Biovolatilization	5 mg L^−1^	4.0%	[207]

* The experimental conditions and the nature of the treated media are shown in Appendix A (Appendix A). UD = unreported data. RDX = hexahydro 1,3,5-trinitro-l,3,5-triazine. TNT = 2,4,6-trinitro-toluene. 2,4-D = 2,4-Dichlorophenoxyacetic acid. 2,4,5-T = 2,4,5-Trichlorophenoxyacetic acid. Orange agent = 2,4-D + 2,4,5-T. TCDD = 2,3,7,8-tetraclorodibenzo-p-dioxina.

## 4. The Mycoremediation of Soils Impacted by War-like Activities

Mycoremediation has proven to be quite efficient in remediating soils impacted by war-like activities since many species of fungi are capable of decomposing primary organic pollutants (explosives and herbicides) or immobilizing inorganic chemicals (toxic metals, toxic metalloids, and radionuclides) derived from these activities. Nevertheless, this bibliographic survey (1980–2023) identified mycoremediation concerning only explosives from soil impacted by wars and the storage/abandonment of ammunition. The interpretation of this gap associated with toxic metals and metalloids, radionuclides, and herbicides is found later in this section.

The analysis of the articles listed in Table 2 points to three primary forms of mycoremediation of soils containing explosives: (i) adding specific fungi to the contaminated soil, (ii) using the soil’s indigenous microbiota, and (iii) composting the contaminated soil. Fungi that degrade explosives are Basidiomycetes, although some Ascomycetes may also present suitable characteristics for performing such degradation.

Adding specific fungi has low efficiency in the degradation of explosives when these pollutants are in high concentrations in soil. In such circumstances, the development of the fungi can decrease significantly. Wood-rotting fungi deserve special attention because they are the most responsible for decomposing TNT in soils as they excrete enzymes capable of destroying aromatic rings. Nevertheless, these fungi’s natural environment (wood) differs significantly from soil. Firstly, soils generally contain less nutrients than wood, these nutrients are in chemically different forms, and soils are spatially much more heterogeneous. Due to this heterogeneity, different soils have substantial disparities in their properties, such as contents and types of organic matter and clay, texture, and microbial population. Moreover, inorganic nutrients necessary for developing wood-rotting fungi are usually in limited quantities in soils. Because of this, it is necessary to add lignocellulosic substrates (corn cobs, wheat or alfalfa straw, wood chips, bark, and peat) to ensure suitable carbon and nitrogen supplies [216]. Since some soils contain unfavorable ratios between carbon and nitrogen, the amount of organic amendments required to allow adequate fungal growth is often high. Consequently, the corresponding costs may make the practice of adding pollutant-decomposing fungi to these types of soils economically uninteresting.

Indigenous microbiota of contaminated soils have tolerant microorganisms, thus allowing for work with higher contamination levels. However, a limitation of using indigenous microbiota is the microbial consortium’s complexity, which will always require specific optimization for each change in the population of microorganisms. Moreover, knowing which fungal species, among several other species, is primarily responsible for decomposing an organic pollutant becomes challenging.

In turn, composting consists of mixing contaminated soils with bulking agents (wood and straw, for example) and organic amendments (cattle manure and vegetable waste, among others) to produce humus-like substances [36,81,217]. The bulking agents ensure adequate porosity, aeration, and nutrition. At the same time, organic amendments maintain the optimum carbon-to-nitrogen ratio [36] for aerobic organisms (bacteria and fungi, mainly). During composting, there is a considerable temperature increase due to microbial catabolic activity, and this heating helps degrade the explosives [218]. As listed in Table 2, composting soils containing explosives has presented good results. However, these procedures can be inconvenient mainly for the following reasons: (i) the need for the excavation and transport of large quantities of soil and (ii) long incubation times (many months). These requirements often result in high costs for composting.

Two works listed in Table 2 presented peculiar aspects that deserve special attention. In this sense, In et al. [219] used sewage sludge for composting soil contaminated with TNT. Sewage sludge has readily degradable organic matter and abundant supplies of nitrogen and phosphorus. After adding this organic amendment to the soil, microbial activity can increase considerably, thus resulting in high rates of mycoremediation and other bioremediation processes. Adding sewage sludge for composting contaminated soil with explosives is promising because this organic amendment is an environmental liability, and it is necessary to find valuable destinations for this material. However, it is necessary to assess sewage sludge for high concentrations of toxic metals (cadmium and lead, for example) that would kill decomposer fungi and other microorganisms and contaminate the composted soil. In turn, Tuomela et al. [220] developed a system capable of operating with 13 tons of soil contaminated with TNT. This characteristic is attractive since most research on soil mycoremediation is limited to the laboratory scale with a few tens of grams of soil.

Some works in Table 2 also refer to soils contaminated with the explosives CL-20 (C_6_H_6_N_12_O_12_, Figure 8) and nitrocellulose (C_6_H_10_O_5n_, Figure 9). CL-20 is a polycyclic nitroamine known as hexanitrohexaazaisowurtzitane, whose first preparation occurred in the USA in 1987. CL-20 is the highest-energy conventional explosive ingredient for military purposes [221]. Because of this, CL-20 finds wide use in military activities. Compared to other nitroamines (RDX and HMX, for example), CL-20 has six N-NO_2_ groups, resulting in more significant heat formation and density [222].

In turn, the primary military application of nitrocellulose is as a propellant to move projectiles at high speed [223].

Regardless of the method chosen for the mycoremediation of soils, it is necessary to evaluate the chemical transformation of the explosives. Total remediation implicates a complete conversion of the explosives to CO_2_, but, as reported in Table 2, intermediary-reduced compounds (diaminonitrotoluenes, for example) are possible. Moreover, the release of CO_2_ does not occur when soil humic organic matter incorporates the explosive’s carbon atoms.

As previously discussed, almost all mycoremediation work refers to masses of a few tens of grams of soil. The challenges for the mycoremediation of several tons (or many square kilometers) of soil impacted by war-like activities are with these environmental matrices’ chemical complexity and heterogeneity. The remarkable pedological differences commonly found in soils within a sizeable polluted area create challenges because they require constant optimization of experimental conditions to ensure the growth of fungi responsible for the remediation of pollutants. Moreover, ex situ mycoremediation techniques present an additional challenge: the high cost of transporting large soil masses. Due to all these difficulties, as far as it is known, no publication has been related to the mycoremediation of several thousands of tons of soil containing toxic metals, toxic metalloids, radionuclides, explosives, and herbicides from war zones, military training areas, and shooting ranges. This significant gap occurs even considering the wide application of fungi to degrade or immobilize all cited pollutants in several other environmental contexts [132,135,172,173,174,190,191,192,193,194,195,196,197,198,199,200,201,202,203,204,205,206,207,208,209,210,211,212,213,214,215,224,225,226,227,228,229]. In this sense, Syngh et al. [206] developed a work in which seven strains of fungi remediated agricultural soils contaminated with arsenic, and the results were promising. Thus, the search of Syngh et al. [206] suggested a marked possibility of achieving the successful mycoremediation of highly As-contaminated soil as a result of World War I (Section 2.1). Like the soils studied by Syngh et al. [206], many of Verdun’s soils, for example, served for agricultural cultivation.

The logistical and operational difficulties of recovering vast soil areas suggest that the mycoremediation of soil containing war-derived pollutants will be more viable if applied subsequently and in smaller areas. In this context, the Yugoslav Civil War provided combat scenarios with areas very restricted concerning uranium, thus facilitating the future employment of fungi to immobilize this pollutant derived from depleted uranium ammunition (Section 2.1). Some species of fungi have very desirable characteristics for decontaminating soil impacted by war-like activities in more restricted areas. As discussed below, mushrooms stand out among these species.

Among fungi, mushrooms deserve special attention for removing metals and metalloids from contaminated soils [132]. These fungi are a specific group of Basidiomycetes, are devoid of chlorophyll, are saprophytic, feed on organic matter, and grow on logs (lignicolous), animal dung (coprophilous), and agricultural waste, among others. Mushrooms can be unicellular or multicellular and reproduce asexually or sexually [107,132].

The annual production of mushrooms worldwide reaches the mark of many millions of tons. This amount of production demonstrates how the supply of mushrooms would be suitable for the mycoremediation of soils impacted by war-like activities. Fungi such as mushrooms have vigorous hyphal growth, ensuring high penetration power in environments such as soil and a high contact area. Combined with these characteristics, the exceptional capacity of these organisms to secrete enzymes capable of degrading complex chemical species opens up a wide range of possibilities for mycoremediation [107].

**Table 2 jof-10-00094-t002:** Works concerning the mycoremediation of soil impacted by explosives after war-like activities.

Pollutant	Fungal Species	Location of the Contaminated Soil Sampling	Main Results	Reference
TNT	*Phanerochaete chrysosporium*	U.S. Army munitions depot at Umatilla (Oregon)	Efficient biotransformation (not mineralization) of TNT at concentrations < 20 ppm	[230]
TNT	*Phanerochaete velutina*	Military storage area in Finland	TNT degradation of 80%	[184]
CL-20	Undefined indigenous species	Soils enriched with CL-20 from different areas of New Jersey (USA)	CL-20 degradation of up to 96%	[231]
TNT, 2,4-DNT and 2,6-DNT	Undefined indigenous species (including fungi)	Inactive munitions plant near Weldon Spring (USA)	High nitroaromatic compound degradation to CO_2_	[232]
TNT	Undefined indigenous species (including fungi)	Inactive munitions plant near Weldon Spring (USA)	Better TNT mineralization (CO_2_ production) under microaerated conditions	[233]
TNT, RDX, and HMX	*Phanerochaete chrysosporium*	Naval weapons station in Yorktown (USA)	Reductions of 98.5%, 70.5%, and 95.8% for TNT, RDX, and HMX, respectively	[234]
TNT	*Phanerochaete chrysosporium*	Agricultural soil from Utah (USA)	TNT degradation of 85%	[175]
TNT	*Mucor sp*. T1-1 and *Aspergillus niger* N2-2	Black and red soils widely spread in western and eastern Georgia (USA)	TNT degradation varied from 79% to nearly 100%	[186]
TNT	Undefined indigenous species (including fungi)	U.S. Army munitions depot near Umatilla (Oregon)	Aqueous supernatants without nitroaromatic compounds	[235]
TNT	*Phanerochaete velutina*	Finnish soils	*Phanerochaete velutina* degraded 70% of TNT on lab and pilot scales	[220]
TNT	Thermophilic microorganisms in compost (including fungi)	Garden soil from Massachusetts (USA)	There was no TNT mineralization, but there was a reduction in aminonitrotolenes	[188]
TNT and RDX	Information not found	Information not found	Total RDX conversion to CO_2_, no evidence for TNT benzene ring breakage, high incorporation of TNT into humic materials	[236]
TNT	Microorganisms in compost (including fungi)	Former ammunition plant Werk Tenne in Germany	In anaerobic conditions followed by an aerobic environment, there was a high decrease in TNT concentration and complete disappearance of the reduced amino-dinitrotoluenes	[237]
TNT, RDX, and HMX	Microorganisms in compost (including fungi)	U.S. Army munitions depot at Umatilla (Oregon)	After composting soils highly contaminated with TNT and RDX, nondetectable levels of these explosives were achieved	[238]
TNT	Microorganisms in compost (including fungi)	U.S. Army munitions depot at Umatilla (Oregon)	No TNT mineralization or formation of amino-dinitrotoluenes was obtained; approximately 70% of carbon was incorporated into the organic fraction	[239]
Nitrocellulose	Microorganisms in compost (including fungi)	U.S. Badger Army Ammunition Plant (Wisconsin)	After composting highly contaminated soils with nitrocellulose, the concentration of this explosive was highly reduced (99.9%)	[240]
TNT	Microorganisms in compost (including fungi)	Former ammunition plant Werk Tenne in Germany	After composting soils contaminated with TNT, this explosive was efficiently degraded (≥99.6%)	[241]
TNT	Microorganisms in compost (including fungi)	Information not found	After composting soils contaminated with TNT, this explosive and its by-products became nonextractable and hydrolyzable	[242]
TNT	Undefined indigenous species (including fungi)	Information not found	Partially reduced TNT was incorporated into the soil with subsequent reduction	[243]
TNT	Autochthonous microorganisms (including fungi)	TNT-contaminated site in the Czech Republic	TNT removal of more than 90%	[244]
TNT	Indigenous microbial consortium (including fungi) in sewage sludge	TNT-contaminated soil from South Korea	TNT removal (with CO_2_ release) between 80% and 85%	[219]
TNT	Indigenous microbial consortium (including fungi)	TNT-contaminated soil from Northwest China	Total organic carbon contents decreased by 48.9% in the liquid phase of a slurry reactor	[245]
TNT	Microorganisms in compost (including fungi)	Aberdeen Proving Ground U.S. Army Installation	After composting soils contaminated with TNT, there was a reduction of this explosive and subsequent binding of the reduced products to soils; however, no TNT mineralization occurred	[246]

Mushrooms may accumulate several elements in their fruit bodies [247,248], and this accumulation ability is species-specific [249]. In this sense, some mushroom species preferably accumulate specific elements, as with *Telephora penicillata*, which assimilate extraordinarily high concentrations of arsenic, cadmium, copper, and zinc. In a Czech forest, this species accumulated 2130, 330, 26, and 4 times more As, Cd, Cu, and Zn than other mushroom species [250]. The very high capacity of *T. penicillata* to accumulate arsenic also reveals a suitable way to decontaminate soils impacted by arsenic-based ammunition. Even with assimilative capacities much lower than arsenic’s, *T. penicillata* has remarkable potential to remediate soils containing copper and zinc derived from the corrosion of ammunition capsules. In addition to the ability of mushrooms to assimilate and bioaccumulate metals and metalloids, these fungi grow relatively quickly and are easy to harvest. Therefore, mushrooms present very desirable characteristics and the potential to decontaminate soils from war zones and shooting range areas. This same argument is valid for radionuclides and herbicide mycoremediation proposals, as will be discussed later.

Fungal species have demonstrated a high capacity for adaption in environments heavily contaminated by radionuclides. From a strictly military point of view, radionuclides come from the radioactive fallout derived from nuclear weapon tests from the 1950s to the 1970s [251]. However, after the Chernobyl accident on 26 April 1986, it became clear that using nuclear fission for peaceful purposes can also enormously pollute the environment. The primary way in which many fungi mitigate radionuclide pollution is by immobilizing these species in their tissues, thus preventing migration to other environmental compartments, such as groundwater and the atmosphere. The fungal species that perform this immobilization very efficiently are saprotrophic fungi, whose hyphae have an enormous surface area [252].

When Chernobyl Reactor 4 exploded in April 1986, local radioactivity exceeded normal levels by five times. Eighteen years after this accident, researchers observed several fungal species with high radionuclide tolerance. The predominance among these tolerant fungi was melanized species (i.e., *Cladosporium* spp.) that presented directional growth or radiotropism [253].

Like other fungal species, mushrooms can also hyperaccumulate radionuclides, as observed for cesium 137, a byproduct of nuclear fission. The species *Gomphidius glutinosus* concentrated cesium 137 by a factor of 10,000× concerning the background contamination level [248,254]. This perspective is very interesting for decontaminating soils impacted by the deposition of radioactive isotopes diffusely (nuclear weapon tests during the Cold War) and punctually (sites of depleted uranium ammunition impacts in the territories from the former Yugoslavia). This ability of mushrooms to assimilate radionuclides from soils is the basis for monitoring food contamination.

As previously stated, before the Chernobyl accident, cesium 134 (half-life 2.06 years) and 137 (half-life 30 years) came from nuclear weapons testing, and analyses of mushrooms were one the most important ways to detect this pollution. After Chernobyl, these radioactive isotopes of cesium also began to be largely monitored in mushrooms in many European countries, such as Austria, Belgium, former Czechoslovakia, Finland, Germany, Italy, Norway, Sweeden, the United Kingdom, and former Yugoslavia. In Japan, 25 edible species of mushrooms, such as *Lentinus edodes*, *Flammulina velutipes*, *Pleurotus ostreatus,* and *Pholiota nameko*, served as biomonitors for the human ingestion of three radioactive isotopes: ^134^Cs, ^137^Cs, and ^40^K [255].

Mushrooms and other species of fungi are highly capable of immobilizing uranium in soils [93]. However, as previously discussed, the chemistry of uranium is very complex, which makes the implementation of chemical and physical remediation processes difficult. On the other hand, microbial remediation processes (including mycoremediation) have many advantages in immobilizing uranium in soils as microorganisms release enzymes able to reduce UO_2_^2+^_(aq)_ to UO_2(s)_. This enzymatic reduction can occur directly or indirectly by capturing electrons from added organic-electron-donating compounds such as lactate and acetate [256].

Nevertheless, there are challenges for successful microbial uranium reduction. The first is to ensure that electron donors’ mass distribution (horizontal and vertical) is uniform on the surface and subsurface of uranium-contaminated soils. In field experiments, this homogeneous distribution is complicated. Another challenge is that adding electron-donating compounds promotes the proliferation of other microorganisms not involved in reducing uranium (VI) to uranium (IV). Moreover, other aspects can affect this reduction. Among these aspects, high concentrations of Ca^2+^, Fe^2+^, and Al^3+^ may decrease the fixation of UO_2_^2+^on phosphates. In turn, if the fixation of UO_2_^2+^ decreases, its toxicity to the microbial population compromises the reduction of UO_2_^2+^_(aq)_ to UO_2(s)_. Furthermore, any modification in soil chemistry that can lower pH and leach UO_2_^2+^ from the solid to the liquid phase of the soils has the potential to inhibit the microbiota responsible for reducing UO_2_^2+^_(aq)_ to UO_2(s)_ [256,257].

Faced with the challenges discussed above and considering that large areas of soil are very heterogeneous chemically and physically, practical results of the in situ bioremediation (including mycoremediation) of soils containing uranium tend to be worse than theoretical expectations. However, for more restricted areas of soil, these challenges decrease considerably. As earlier stated in this section, such areas of localized uranium contamination in soils are common in many territories of the former Yugoslavia, which received DU ammunition impacts. Therefore, mycoremediation can be a promising alternative for decontaminating soils impacted by the fighting during the Yugoslav Civil War.

On 29 December 2022, the government of the United States of America announced that it would invest USD 29 million to decontaminate a Vietnamese area (including soil) containing dioxin. This area was home to a large American air base during the Vietnam War, where planes filled with Agent Orange took off to spray this herbicide over Vietnamese forests, soils, and rivers [258]. However, this decontamination proposed by the U.S. government is based on the incineration of soils at high temperatures to transform the dioxin into harmless compounds. In addition to being expensive, this type of decontamination would destroy the cultivation capacity of thousands of tons of soil [259].

Like discussions for other soil pollutants from war-like activities, mycoremediation does not appear among the procedures for decontaminating Vietnamese soils containing Agent Orange. Nguyen et al. [199] isolated some fungi from Vietnamese soils contaminated with 2,4-D and 2,4,5-T. Subsequently, they used two filamentous fungi to decompose both herbicides in culture media, achieving promising outcomes. Nevertheless, large-scale experiments using fungi directly in soils from war zones containing 2,4-D and 2,3,5-T do not exist.

Anasonye et al. [260] employed mycoremediation of Finnish soils contaminated with polychlorinated dibenzo-*p*-dioxins and dibenzofurans. The authors found that bioaugmentation of the fungi *Stropharia rugosoannulata* or *Phanerochaete velutina* could degrade 62 to 64% of the pollutants. Although this experiment demonstrated the potential of mycoremediation on a small scale (350 g of soil), results for pilot and field scales can vary greatly, and research needs to continue in this direction. To our knowledge, there is no field-scale mycoremediation work to remove dioxins from soils affected by Agent Orange in Vietnam.

## 5. Conclusions and Perspectives

Analyzing the articles in this literature review makes it possible to highlight three key aspects that can lead to the successful mycoremediation of soils (including those impacted by war-like activities):(a)Fungal adaptability to the contaminated environment: Fungi can adapt and survive in soils contaminated with explosives, metals, metalloids, radionuclides, and herbicides, provided the conditions are not highly adverse.(b)Mycelial growth: The mycelium (network of hyphae) of filamentous fungi efficiently explores soils and substrates for degrading nutrients and compounds. This morphological characteristic helps increase the interaction area between fungi and organic war pollutants (explosives and herbicides), facilitating degradation. Moreover, this vast contact area with metals, metalloids, and radionuclides enhances their adsorption.(c)Syntropy: Filamentous fungi can form symbiotic relationships with other microbial species, such as bacteria, in a process known as syntropy. This microbial cooperation can enhance the effectiveness of explosive and herbicide degradation as different microorganisms can play complementary roles in compound transformation. This same syntropic effect can also magnify the adsorption of metal and metalloid ions and radionuclides.

Despite the key aspects discussed above, using mycoremediation to recover soils with toxic metals and metalloids, radionuclides, and herbicides from war zones presents an immense gap in the publications. The same statement is valid for soils containing explosives and toxic metals (notably lead) from military training areas and shooting ranges. To a large extent, this gap exists due to the remarkable soil chemical complexity and heterogeneity and the logistical and economic challenges of removing and transporting several thousands of tons of contaminated soil, considering ex situ mycoremediation methods. A less limiting situation would be mycoremediation in the in situ modality, in which soils would receive non-indigenous fungi or undergo the stimulation of indigenous fungal populations. Nevertheless, paying particular attention to soil heterogeneity is necessary because the heterogeneous soil environment requires optimizations for each polluted area to ensure the suitable development of added or stimulated fungi.

Despite the challenges previously discussed, mycoremediation has two significant aspects capable of supporting future intensifications of its use to recover soils impacted by war-like activities. The first of these aspects is the vast proof of the effectiveness of fungi (notably filamentous fungi) for degrading or immobilizing the five classes of warfare pollutants considered in this manuscript. The second aspect is that mycoremediation does not destroy the structure of contaminated soils as incineration does. This feature is relevant given the growing demand for food and arable land.

## Figures and Tables

**Figure 1 jof-10-00094-f001:**
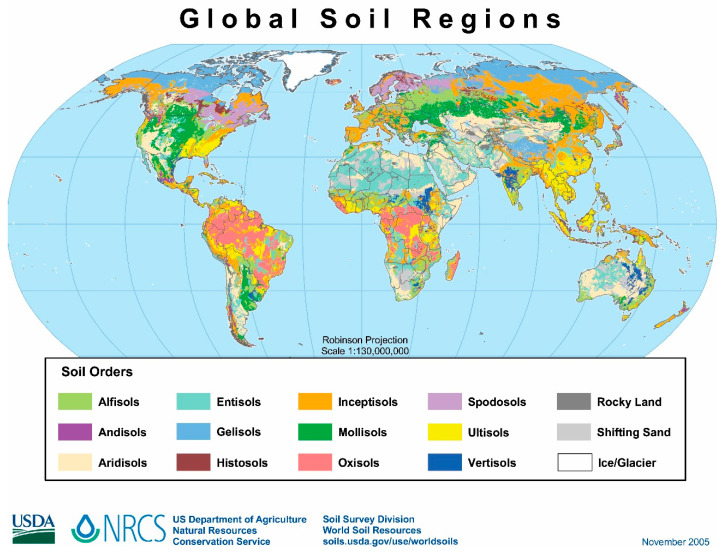
Dominant soils of the world. https://upload.wikimedia.org/wikipedia/commons/e/e5/Global_soils_map_USDA.jpg, accessed on 7 August 2023).

**Figure 2 jof-10-00094-f002:**
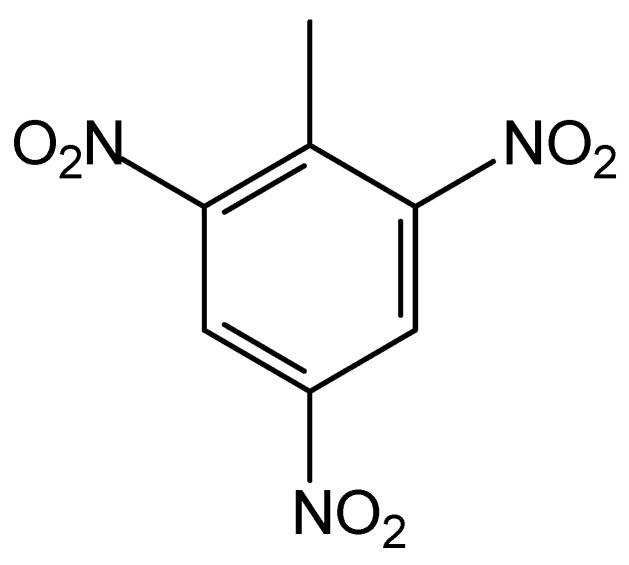
The structural formula of the TNT explosive.

**Figure 3 jof-10-00094-f003:**
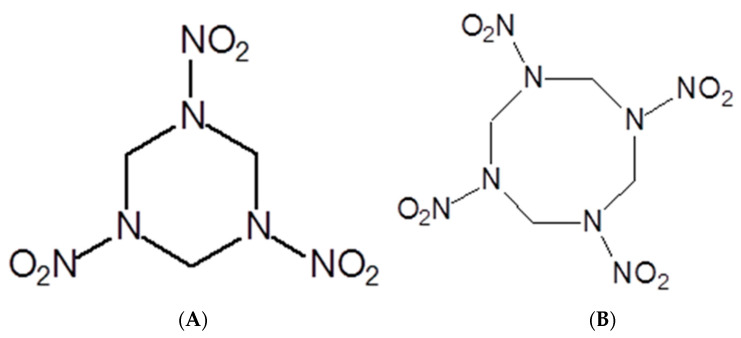
Structural formulas of RDX (**A**) and HMX (**B**) explosives.

**Figure 4 jof-10-00094-f004:**
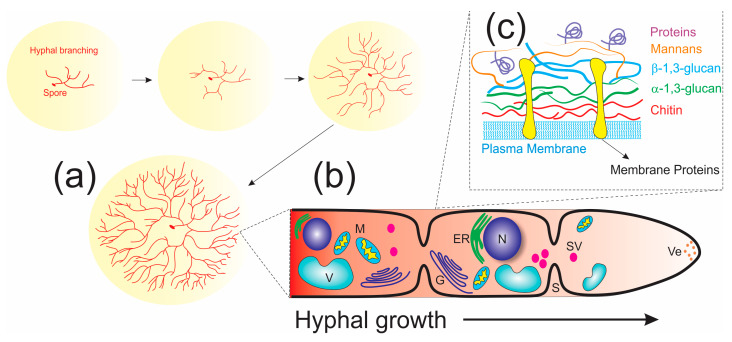
Diagrammatic representation of fungal cellular organization and growth: (**a**) Spores germinating to give rise to long cylindrical cells known as hyphae that produce the mycelium network. (**b**) Main components of the fungal cell: N = nucleus, M = mitochondria, V = vacuole, Ve = vesicle, SV = secretory vesicle, G = Golgi system, S = septum. (**c**) Architecture of the fungal cell wall.

**Figure 5 jof-10-00094-f005:**
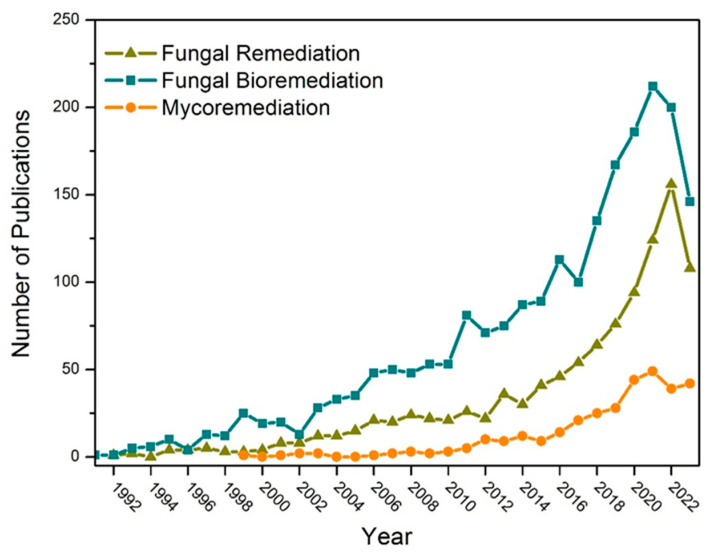
Representation of publication trends for “fungal remediation”, “fungal bioremediation”, and “mycoremediation” between 1990 and 2023 by Web of Science (www.webofscience.com/ (accessed on 30 October 2023).

**Figure 6 jof-10-00094-f006:**
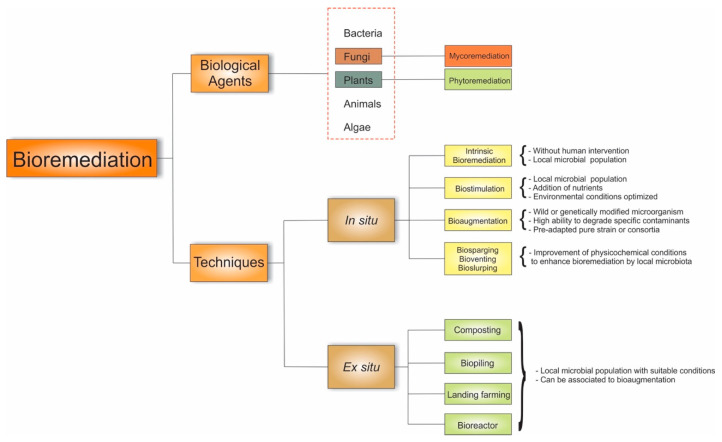
Mycoremediation techniques to remove pollutants from contaminated environments.

**Figure 7 jof-10-00094-f007:**
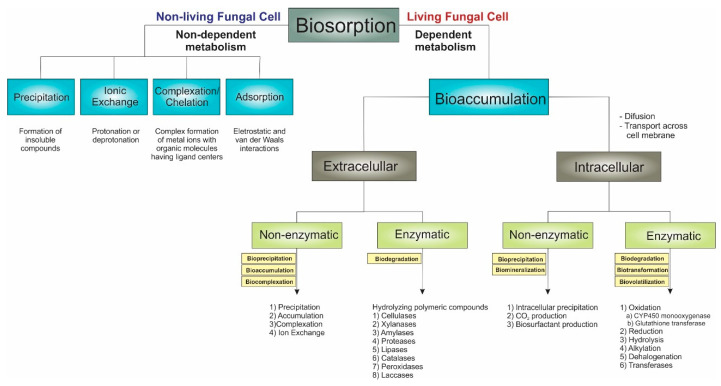
Adopted mechanisms of mycoremediation of environmental pollutants (organic compounds, radionuclides, and toxic metals and metalloids).

**Figure 8 jof-10-00094-f008:**
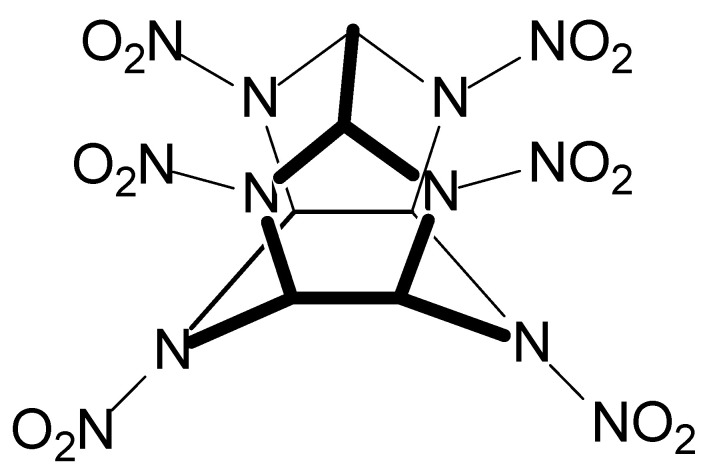
The structural formula of the CL-20 explosive.

**Figure 9 jof-10-00094-f009:**
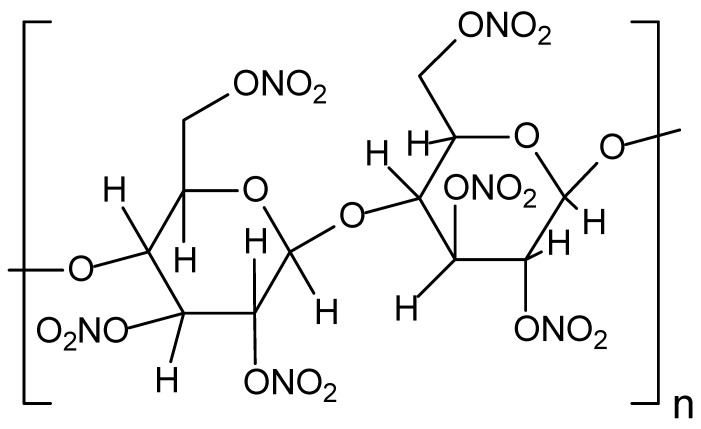
The structural formula of the nitrocellulose explosive.

## Data Availability

No new data were created or analyzed in this study. Data sharing is not applicable to this article.

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
