# Peer review of "A Review about the Mycoremediation of Soil Impacted by War-like Activities: Challenges and Gaps"

_jof, 2024, doi:10.3390/jof10020094_

Round 1

Reviewer 1 Report

Comments and Suggestions for Authors

In this review, the mycoremediation had been summarized for their successful applications in removing all pollutants from warlike activities. However, the mycoremediation of soils in former war zones and those impacted by military training and shooting ranges was still very incipient, with most applications only emphasize on explosives. The information provided is important for successful mycoremediation of soils (including those impacted by warlike activities) that contaminated with explosives, metals, metalloids, radionuclides, and herbicides. Therefore, I may recommend publishing this manuscript after revisions are achieved.

Detailed comments are as below:

1. Scientific writing of the whole manuscript need improve. Carefully check the writing and grammar throughout the text, including table and figure captions. For example, there was no period after the environmental point of view in line 57 and in Germany, Australia, and Asia [109] in line 533 .

2. The annotated references were not in the correct format and should be superscript, and in line 642, [133 the trailing half parenthesis was missing.

3. All font sizes should be the same in the text. Paragraph 6 under "3.4" was larger than the rest of the text.

4. In "3.2.1. Wars: from World War I to the Yugoslav Civil War", the course of the war was overstated and should be briefly described, with emphasis on the environmental impact of war activities on the soil.

5.In "3.4.2. Fungi and Their Action as Bioremediation Agents", the role of fungal remediation should be supplemented.

6. Conclusion should be shortened and more concise.

7. Check the reference list. For example, “Harari, Y.N. Sapiens: A Brief History of Humankind; Harper Perennial: 2018. 'in line 1205. not complete.

Author Response

Reviewer 1

Comment 1Scientific writing of the whole manuscript need improve. Carefully check the writing and gramar throughout the text, including table and figure captions. For example, there was no period after “the environmental point of view” in line 57 and “in Germany, Australia, and Asia [109]” in line 533.

Reply to comment 1 – The entire manuscript was read rigorously, including the captions to tables and figures. In this sense, the first example cited by Reviewer 1 (line 57) was corrected, while the second example was removed from the text due to a decrease in the manuscript's size.

Comment 2The annotated references were not in the correct format and should be superscript, and in line 642, “[133” the trailing half parenthesis was missing.

Reply to comment 2 – The citation of references throughout the manuscript (in the text and the References section) followed the recommendations in the Guide for Authors. The error concerning reference 133 (122 in the new version of the manuscript) was corrected (line 501).

Comment 3All font sizes should be the same in the text. Paragraph 6 under “3.4” was larger than the rest of the text.

Reply to comment 3 – The text was corrected, and all inconsistencies found regarding font size were corrected.

Comment 4In “3.2.1. Wars: from World War I to the Yugoslav Civil War”, the course of the war was overstated and should be briefly described, with emphasis on the environmental impact of war activities on the soil.

Reply to comment 4 – Comments on the course of the wars were removed, and only discussions strictly referring to the environmental impacts of wars on soil were maintained.

Comment 5In “3.4.2. Fungi and Their Action as Bioremediation Agents”, the role of fungal remediation should be supplemented.

Reply to comment 5 – In fact, the former section 3.1 (Fungi and Their Action as Bioremediation Agents) alluded to the forms of action of fungi in remediation processes. However, more detailed information about these processes was in the former section 3.3 (Mycoremediation Mechanisms). In this new version of the manuscript, the former sections 3.1 and 3.2 have become a single section (Section 3.1) entitled Mycoremediation and its Techniques (Line 502). This new section defines Mycoremediation, locates it within the larger universe of bioremediation, and discusses the different ways of performing it. The section Mycoremediation Mechanisms is now in section 3.2 (Line 631).

Comment 6 – Conclusion should be shortened and more concise.

Reply to comment 6 – Parts of the "key aspects" were excluded in this section. In this sense, the former item c) was excluded because it was redundant. Moreover, the remaining items had their texts shortened.

Comment 7 Check the reference list. For example, “Harari, Y.N. Sapiens: A Brief History of Humankind; Harper Perennial: 2018” in line 1205 is not complete.

Reply to comment 7 - The entire list of references has been reviewed, and formatting inconsistencies have been corrected. The corrected parts are highlighted in red.

Reviewer 2 Report

Comments and Suggestions for Authors

The review is very comprehensive and scientifically interesting to read. However, in my opinion the manuscript is too long and requires some heavy trimming, as in it's present form it reads more like a book chapter than a journal review. My suggestion to synthesize sections from 1 to 3.3 into one or two small introductory sections stating the problem with some examples, and then focusing most of the review on the mycoremediation aspect. In my opinion there are also far too many figures - again, not a negative point if you are writing a chapter of a book, but too many for a journal review (for instance, I don't think we need a pic of a nuclear explosion). 

Comments on the Quality of English Language

English is good. 

Author Response

Reviewer 2

Comment 1 - The review is very comprehensive and scientifically interesting to read. However, in my opinion the manuscript is too long and requires some heavy trimming, as in it's present form it reads more like a book chapter than a journal review. My suggestion to synthesize sections from 1 to 3.3 into one or two small introductory sections stating the problem with some examples, and then focusing most of the review on the mycoremediation aspect. In my opinion there are also far too many figures - again, not a negative point if you are writing a chapter of a book, but too many for a journal review (for instance, I don't think we need a pic of a nuclear explosion.

Reply to comment 1 - The size of the manuscript has undergone a significant reduction. The total number of pages decreased from 69 to 51, and the number of words from 26,800 to 21,493. In this text size reduction, section 2, which described how this literature review was carried out, was excluded, and the text was transferred to section 1 (Text highlighted in green, lines 86-101). Another part of the former text was also inserted in the Introduction (Text highlighted in green, lines 70 to 79). The section that addressed the formation and characteristics of soils was excluded. The new Theoretical Foundation is divided into two parts: 2.1 How have World War I, World War II, the Cold War, the Vietnam War, and the Yugoslav Civil War environmentally impacted soils? (Part highlighted in red, lines 103-104), and 2.2 How do military training and shooting ranges environmentally impact the soils? (Part highlighted in red, line 399). Regarding the first version of the manuscript, 12 figures were removed, including the figure (nuclear detonation) highlighted by Reviewer 2.

Reviewer 3 Report

Comments and Suggestions for Authors

The document is long. Some information is not necessary; for example 3.1. Soil: A Very Complex Environmental Matrix, 3.1.1. Physical And Chemical Soil Features. Similarly, there are too many figures which are not relevant to the document. There are nice figures, but these are not informative and do not give strength to the document. 

Table 1 is long and is not discussed properly in the text. 

The section on perspectives should be strengthened. Gaps were well mentioned but challenges are not well mentioned. 

Please try to link sections. 

Some aspects to review in the PDF-reviewed document. 

Author Response

Reviewer 3

Comment 1 - The document is long. Some information is not necessary; for example 3.1. Soil: A Very Complex Environmental Matrix, 3.1.1. Physical And Chemical Soil Features. Similarly, there are too many figures which are not relevant to the document. There are nice figures, but these are not informative and do not give strength to the document.

Reply to comment 1 - The response to this first comment from Reviewer 3 was already included in the response to Reviewer 2's comment.

Comment 2 - Table 1 is long and is not discussed properly in the text.

Reply to comment 2 - Table 1 was simplified and reduced in size by removing the Experimental Conditions column. The information concerning the Experimental Conditions column is in Table S1 of the Supplementary Materials. The discussion about Table 1 was inserted in the text (Part highlighted in red, lines 756 to 765).

Comment 3 - The section on perspectives should be strengthened. Gaps were well mentioned but challenges are not well mentioned.

Reply to comment 3 - Reviewer 1 requested that the Conclusions and Perspectives be shortened. Therefore, to meet Reviewer 3's request to improve the discussion on the challenges concerning the mycoremediation of soil impacted by warlike activities, modifications were made in section 4 (Parts highlighted in red, lines 817-821, 825-827, 873-884). Phrases were also added in the Conclusions and Perspectives section (Parts highlighted in red, lines 1035-1053) to highlight better the challenges of using mycoremediation on soil impacted by warlike activities.

Comment 4 - Please try to link sections.

Reply to comment 4 - The improvement of the connection between the different parts of the text was carried out (Parts highlighted in blue, lines 114-115, 213-217, 303, 312-313, 317-319, 345, 394-398, 444-446, 627-630, 898-900).

 Comment 5 - Some aspects to review in the PDF-reviewed document.

Reply to comment 5 - Reviewer 3 attached a PDF file containing some change requests. Some changes could not be made because the texts in which they were do not exist in the new version of the manuscript. All of the remaining requests were met (Parts highlighted in red, lines 65, 75, 169, 176, 360, 450, 497-501, 545-546, 559, 565, 572, 576, 633, 635, 657, 684, 711, 719, 733-736, 748-749, 793-795, 797, 865-866, 881 (The part as far as it is known) and 1042.

Reviewer 3 requested that the word physicochemical be replaced by the words physical or chemical in some parts of the text and a figure. However, I request permission for the word physicochemical to be maintained as it is one of the most commonly used terms in the literature to allude to the set of properties, such as pH, temperature, aeration, and concentration of chemical species (such as nutrients), necessary to describe the status of several experimental systems, such as biological mycoremediation systems. In this sense, the word physicochemical presents a connotation consistent with the adequacy of experimental conditions for optimizing mycoremediation processes.

There was a request not to leave empty spaces near Tables 1 and 2. Following the formatting of the template received, although more text was added before Table 1 to describe it appropriately, there is still space left. The space left before Table 2 was considerably reduced.

As requested by Reviewer 3, there is now a space between the percentage sign (%) and the numbers.

All parts identified as out of context or unnecessary were removed.

Round 2

Reviewer 2 Report

Comments and Suggestions for Authors

The revisions of the manuscript are welcomed. However, in my opinion this manuscript is still too long to be a review in a journal rather than a chapter in a book. 

My suggestion would be for the authors to further synthesize the information regarding the description of war-time and peace-time activities. For instance, in my opinion section 2 could be summarized into 2 to 3 pages briefly describing the compounds that are accumulated due to military activity, while all the information about what compound were accumulated where and when could be summarized in a table. 

Comments on the Quality of English Language

No problems with the english aside from minor corrections. 

Author Response

Thanks for the valuable comments regarding the manuscript A Review About the Mycoremediation of Soil Impacted by Warlike Activities: Challenges and Gaps (jof2752737). Responses to your comments are below.

Reviewer 2

Comment 1The revisions of the manuscript are welcomed. However, in my opinion this manuscript is still too long to be a review in a journal rather than a chapter in a book. 

My suggestion would be for the authors to further synthesize the information regarding the description of war-time and peace-time activities. For instance, in my opinion section 2 could be summarized into 2 to 3 pages briefly describing the compounds that are accumulated due to military activity, while all the information about what compound were accumulated where and when could be summarized in a table. 

Reply to comment 1 – Compared to the first revised version, this second revised version had its number of pages reduced from 51 (21,786 words) to 46 (19,065 words). As requested by Reviewer 2, the size of Section 2 has been reduced to three pages (half of page 3, entire pages 4 and 5, and half of page 6). To this end, three figures were removed, and parts of the discussion that were not essential for the review theme were removed from the text. Information regarding the description of war-time and peace-time activities was very synthesized. In Section 3.2, three subsections were added to improve the text organization. The titles of these subsections are highlighted in bold red (Lines 446, 474 and 500). 

The toxicological effects of arsenic, copper, iron, lead, and zinc, as well as the explosives TNT, RDX, and HMX, on humans, are abundantly reported in the literature. In this sense, they were not summarized in a table. Instead, the description of these toxicological effects was very summarized in the text of Section 2, with the respective references for more detailed information. The parts of Section 2 relating to summary information on toxicological effects are highlighted in bold red (Lines 123-125, 141-142, 156-158, 191-192). Still, in Section 2, discussions about the pollutant concentrations in the soil impacted by warlike activities were summarized. These summarized parts are highlighted in bold red (Lines 107-109, 113-116, 198-200, 202-204).

Reviewer 3 Report

Comments and Suggestions for Authors

The authors have made the corrections solicited. 

Author Response

Thank you very much for your valuable recommendations.